# Challenges in the Highly Selective [3 + 1]-Cycloaddition of an Enoldiazoacetamide to Form a Donor–Acceptor *Cis*-Cyclobutenecarboxamide [note 1]

**DOI:** 10.3390/molecules26123520

**Published:** 2021-06-09

**Authors:** Sipak Joyasawal, Donghui Ma, Michael P. Doyle

**Affiliations:** Department of Chemistry, The University of Texas at San Antonio One UTSA Circle, San Antonio, TX 78249, USA; sipak.joyasawal@utsa.edu (S.J.); ma698@purdue.edu (D.M.)

**Keywords:** donor–acceptor cyclobutenecarboxamide, [3 + 1]-cycloaddition reaction, 3-siloxy-2-diazo-3-butenamides, *Z*-γ-substituted metallo-enolcarbene, box ligands, copper(I)

## Abstract

A substituted donor–acceptor cyclobutenecarboxamide is synthesized with modest enantiocontrol through a chiral copper(I) complex catalyzed [3 + 1]-cycloaddition reaction of α-acyl diphenylsulfur ylides with 3-siloxy-2-diazo-3-butenamides. With a methyl substituent on the 4-position of the 3-butenamide, the *cis*-vicinal-3,4-disubstituted cyclobutenecarboxamide is formed with >20:1 diastereocontrol. Donor-acceptor 3-methyl-2-siloxycyclopropenecarboxamide is rapidly formed from the reactant enoldiazoamide and undergoes catalytic ring opening to give only the *Z*-γ-substituted metallo-enolcarbene. Elimination from 3-siloxy-2-diazo-3-pentenamide to form the conjugated 3-siloxy-2,4-pentadienamide is competitive but minimized at low temperature.

## 1. Introduction

The four-membered carbon ring is an important structural framework present in natural products and biologically active compounds, but is less accessible than are other ring structures [1,2,3]. Furthermore, cyclobutanes and cyclobutenes are integral to synthetic strategies involving facile ring-expansion or ring-cleavage reactions [4,5]. The synthesis of cyclobutenes is normally relegated to [2 + 2]-cycloaddition reactions between alkynes and activated alkenes [6,7,8,9,10,11,12,13,14,15], and diastereocontrol for the synthesis of 3,4-disubstituted cyclobutenes is controlled by the geometrical isomerism of the reactant alkyne. Until recently [16,17], however, there were no examples of a broadly applicable enantioselective version of this [2 + 2]-cycloaddition process. Diastereocontrol is conveniently established by the alkene reactant with this methodology; enantiocontrol is the challenge (Figure 1a).

Previous research from our laboratory established that catalytic [3 + 1]-cycloaddition of silyl group-protected enoldiazoacetate esters with α-acyl dimethylsulfur ylides was effective in forming stable donor−acceptor cyclobutene derivatives (Figure 1b) [18]. This methodology produced 1,2,4-trisubstituted (R^1^ = H) 2-siloxycyclobutenecarboxylates in good yields with uniformly high enantiocontrol. Diastereoselectivity for the formation of 1,2,3,4-tetrasubstituted (R^1^ = Me) 2-siloxycyclobutenecarboxylates was good (>10:1) to excellent (>20:1) with the *trans* isomer dominant; however, even in this earlier study there were indications that formation of the *cis*-diastereomer could be competitive.

Prior research has suggested that diazoamides are more stable and more selective in their catalytic reactions emanating from metal carbene intermediates [19,20], and we anticipated that this selectivity could be applied to catalytic [3 + 1]-cycloaddition of silyl group protected enoldiazoacetamides. However, initial efforts indicated that the same conditions and catalysts that were effective with enoldiazoacetates were not as productive or selective with enoldiazoacetamides. In particular, reactions with α-acyl dimethylsulfur ylides gave low yields for the cycloaddition product, and stereoselectivities were low. To enhance both the efficiency of the transformation and its selectivity to produce substituted donor–acceptor cyclobutenecarboxamides with exceptional stereocontrol, we undertook a comprehensive effort to optimize reactants, conditions, and catalyst ligands to achieve high yields, as well as high enantioselectivities and diastereocontrol (Figure 1c).

## 2. Results and Discussion

We began our investigation with the cycloaddition of TIPS-protected *N,N*-dimethyl-enoldiazoacetamide **1a** with α-benzoyl dimethylsulfur ylide **2a** using the same copper(I) catalyst and chiral ligand (**L1**, Scheme 1) that were most effective in reactions with enoldiazoacetates (Figure 2) [18]. However, reaction at room temperature under the same conditions produced the [3 + 1]-cycloaddition product **3a** in 58% yield having 0% ee after complete dinitrogen extrusion of **1a** (Table 1, entry 1). Since phenyl in place of methyl increases the reactivity of the sulfur ylide [21], α-benzoyl diphenylsulfur ylide **2b** was prepared [22,23,24,25], and its reaction with **1a** under the same conditions gave **3a** in 85% yield with 17% ee (Appendix A, entry 2, Appendix A). The major enantiomer was assigned to be *R* based on its correlation in sign of rotation and relative retention volume by HPLC compared with the [3 + 1]-cycloaddition product with the corresponding enoldiazoacetate [18].

### 2.1. Ligand Control of Enantioselectivity in the [3 + 1]-Cycloaddition of 2-Diazo-N,N-dimethyl-3-(triisopropylsiloxy)but-3-enamide

Using the 4-phenyl-Sabox **L1**, the ester analog of **1a**, methyl enoldiazoacetate, was able to reach 83% ee in its [3 + 1]-cycloaddition reaction with **2b**. That 17% ee (Appendix A, entry 2) could be achieved in reactions with *N,N*-dimethyl-enoldiazoacetamide **1a** was surprising and not easily explained as due to the size of dimethylamido relative to methoxy groups, nor by the expected electronic influence of amide relative to ester groups on the diazo carbon. Consequently, we directed our attention to expanding our search for ligands that might increase enantioselectivity with a survey of Box (bis-oxazoline) and SaBox (sidearmed bis-oxazoline) ligands (Scheme 1). 

To determine the ability of ligands to enhance enantiocontrol in the [3 + 1]-cycloaddition reactions of *N,N*-dimethyl-enoldiazoacetamide **1a** with ylides **2a** or **2b** using CuOTf·Tol_1/2_, the series of Box and Sabox ligands (**L2**–**L17** in Scheme 1) were screened (Table 1 and Appendix A). Low enantioselectivities (~5% ee) and product yields were achieved in copper(I) catalyzed reactions with the dimethylsulfur ylide **2a** using **L3**, **L12**, and **L17** (55–60% yield, 5% ee, Appendix A, entries 5, 12, 18). However, changing to the α-benzoyl diphenylsulfur ylide (**2b**) brought about an increase in enantioselectivity (34% ee) using Box **L2** (Appendix A, entry 3), which was greater than that with Sabox **L1**, but Sabox **L5** (Table 1, entry 2) with the 3,5-di-*tert*-butylbenzyl sidearm gave a near doubling in % ee. Use of indanyl Sabox ligands **L6**–**L8** (Appendix A, entries 7, 8, 9) were disappointing with ee’s less than 15%, but **L9**, and **L10** provided modest enantiocontrol (Table 1, entries 3, 4). Only with the *cis*-3,4-diphenyl Sabox ligands did enantiocontrol reach beyond 60% ee, and **L17** (Table 1, entry 10) was designated to be the optimum ligand. Lowering the temperature increased the ee by 4% for each 20-degree decrease (Table 1, entries 10, 11, 12). The use of CuBF_4_(CH_3_CN)_4_ in place of CuOTf·Tol_1/2_ had no meaningful influence on % yield or enantioselectivity (Table 1, entries 13, 14, 15). Use of alternative solvents, including DCE, toluene, and ethyl ether, with **L17** led to much lower % ee values (see Appendix A).

### 2.2. Ligand Control of Diastereoselectivity and Enantioselectivity in the [3 + 1]-Cycloaddition of 2-Diazo-N,N-dimethyl-3-(triisopropylsiloxy)pent-3-enamide

With moderate enantioselectivity in place for the [3 + 1]-cycloaddition of *N,N*-dimethyl-enoldiazoacetamide **1a** with α-benzoyl diphenylsulfur ylide **2b**, our next goal was controlling diastereoselectivity. Use of the γ-methyl-substituted **1b** was anticipated to result in the formation of vicinal 3,4-disubstituted donor-acceptor cyclobutene **3b** which can be produced as either the *cis* (***Z*-3b**) or *trans* (***E*-3b**) diastereomer [26,27,28,29]. Prior results from the corresponding γ-methyl-enoldiazoacetate showed a substantial increase in enantioselectivity (to 95% ee) over reactions with the enoldiazoacetate without γ-substitution (83% ee), but a reduction in the dr (diastereomeric ratio) from *t*:*c* > 20:1 with **2a** to *t*:*c* = 13:1 with **2b** [18] so we anticipated an increase in enantioselectivity but a further decrease in the *t*:*c* ratio in reactions of **1b** with **2b**. γ-Methyl-substituted **1b** prepared by TIPS transfer from TIPSOTf to the precursor α-diazo-β-ketoacetamide promoted by triethylamine yielded **1b** (Scheme 2) having a 3:1 *E:Z* ratio [30]; the expected [3 + 1]-cycloaddition reaction with this stereomeric mixture catalyzed by CuOTf·Tol_1/2_/**L17** produced both ***Z*-3b** and ***E*-3b** in a 7:1 ratio (Appendix A, entry 1, Appendix A) for a reversal in diastereocontrol but in low yield (35%), so we surveyed ligands again and found that **L14** gave the highest yield of ***Z*-3b** (57%) with a ***Z*/*E*-3b** ratio of 6 (64% ee):1(7% ee) along with the surprising production of diene **4** (22% yield) for an overall yield of 89% (Scheme 3). Recognizing that the individual isomers of **1b** could be acting differently in this catalytic reaction, we prepared ***Z***-**1b** (25:1 ***Z***:***E***, Scheme 4) [30] and subjected this diazo compound to the same reaction conditions and found both ***Z*-3b** and ***E*-3b** in a 9:1 ratio (77% yield with 68% ee for ***Z*-3b** and 20% ee for ***E*-3b**), along with **4** (15% yield). Since the donor-acceptor cyclopropene formed from the reactant enoldiazoacetamide **1b** could be the resting state for the intermediate metallo-enolcarbene [31], we prepared cyclopropene **5** (Scheme 5) **[32]** and subjected this compound to the same reaction conditions to find both ***Z*-3b** and ***E*-3b** in a 12:1 ratio (73% yield with 71% ee for ***Z*-3b** and 16% ee for ***E*-3b**), along with **4** (8% yield). The same reaction performed at −20 °C showed much higher selectivity overall (Scheme 3), with an impressive 24:1 ***Z*-3b**:***E*-3b** ratio and 79% ee for the *cis* diastereoisomer, and significantly less diene (Scheme 3).

### 2.3. Ligand Control of Enantioselectivity in the [3 + 1]-Cycloaddition of 2-Diazo-N,N-dimethyl-3-methyl-2-(triisopropylsiloxy)-1-cyclopropenecarboxamide

The discovery that there was such a substantial reverse in diastereoselectivity in reactions performed with the enoldiazoacetamide (**1b**) from that found with the corresponding enoldiazoacetate was surprising and prompted investigation of the effect of the ligand **L** on diastereocontrol. To avoid perturbations in selectivities arising from the different reactivities and selectivities of ***E*-** and ***Z*-1b**, we chose to use donor-acceptor cyclopropene **5** as the carbene source. Previous results have shown that the donor-acceptor cyclopropene undergoes ring opening with transition metal catalysts to form only the *Z*-metallo-enolcarbene isomer [31]. We surveyed a representative series of Box and Sabox ligands for [3 + 1]-cycloaddition with **2b**, and these results are given in Table 2. As previously stated, yields of [3 + 1]-cycloaddition products are limited by the competing formation of diene **4**. However, even with this limitation, **L14** (Table 2, entry 6) produced ***Z*-3b** with a 12:1 ***Z****:**E*** ratio at room temperature and allowed the isolation of pure ***Z*-3b** in 67% yield with 71% ee, and a higher yield, dr and % ee were achieved when the reaction was performed at −20 °C (Table 2, entry 7).

## 3. Materials and Methods

### 3.1. General Information

All reactions, unless noted, were performed in oven-dried (120 °C) glassware with magnetic stirring under an inert atmosphere of dry nitrogen. Analytical thin layer chromatography was carried out using EM Science silica gel 60 F254 plates (MilliporeSigma, Burlington, MA, USA); visualization was accomplished with UV light (254 nm). Column chromatography was performed on CombiFlash^®^ Rf200 and Rf+ purification systems (Teledyne Technologies, Thousand Oaks, CA, USA) using normal phase disposable columns. ^1^H-NMR spectra were recorded on a Bruker spectrometer (500 MHz, Bruker Corporation, Billerica, MA, USA). Chemical shifts were reported in ppm downfield from tetramethylsilane with the solvent resonance as the internal standard (CDCl_3_, δ = 7.28). Spectra were reported as follows: chemical shift (δ ppm), multiplicity (br = broad singlet, s = singlet, d = doublet, t = triplet, q = quartet, m = multiplet, comp = composite of magnetically non-equivalent protons, dd = doublet of doublets, td = triplet of doublets; dt = doublet of triplets), coupling constants (Hz), integration and assignment. ^13^C-NMR spectra were collected on Bruker instrument (125 MHz, Bruker Corporation) with complete proton decoupling. Chemical shifts are reported in ppm from the tetramethylsilane with the solvent resonance as internal standard (CDCl_3_, δ = 77.0). High-resolution mass spectra (HRMS) were performed on a Bruker MicroTOF-ESI mass spectrometer (Bruker Corporation) with an ESI resource using CsI or LTQ ESI positive ion calibration solution as the standard. Enantioselectivities were determined by HPLC analysis at 25 °C using an Agilent 1260 Infinity HPLC System (Agilent-Technologies, Santa Clara, CA, USA) equipped with an G1311B quaternary pump, G1315D diode array detector, G1329B auto-sampler, G1316A thermostated column compartment and G1170A valve drive. For instrument control and data processing, Agilent OpenLAB CDS ChemStation Edition (1200-series) for LC & LC/MS Systems (Rev. C.01.07 [26]) software was used. Chiralpak OD-H (0.46 mm × 250 mm) columns were obtained from Daicel Chiral Technologies (Chiral Technologies Inc., West Chester, PA, USA). Tetrahydrofuran, dichloromethane, chloroform, and toluene were purified using a JC Meyer solvent purification system. All other solvents were purified and dried using standard methods.

### 3.2. Abbreviations

EtOAc—ethyl acetate, THF—tetrahydrofuran, MeOH—methanol, DCM—dichloromethane, TEA—triethylamine, MeCN—acetonitrile, *i*-PrOH—2-propanol, TLC—thin layer chromatography, TMS—tetramethylsilane (Purchased from Fisher Scientific, Waltham, MA, USA).

### 3.3. Materials

**1a**, **2b** and **5** [18,30,32,33,34], ligands [35,36] and sulfur ylides [22,23,24,25] were prepared by reported methods. All commercially available reagents were used without further purification unless otherwise noted. Preparation of racemic-**3a** was discussed in Figure 2. Compound **1a** was prepared from known diazoketone (**6**) using TIPSOTf and Et_3_N in DCM [18]. Compound **8** was made from known compound *N*,*N*-dimethyl-3-oxopentanamide **7** using *p*-ABSA in acetonitrile [35] as shown in Scheme 6.

### 3.4. General Procedure for Asymmetric Catalytic [3 + 1]-Cycloaddition to Prepare (R)-4-Benzoyl-N,N-dimethyl-2-(triisopropylsiloxy)cyclobut-1-ene-1-Carboxamide (***3a***)

To an oven-dried sealable 2-dram vial equipped with a stir bar were added CuOTf·Tol_1/2_ (12.93 mg 0.05 mmol, 5 mol%) and **L17** (51.84 mg, 0.06 mmol, 6 mmol%). After the vial was evacuated and backfilled with N_2_ three times, dry DCM (1.0 mL) was added via a syringe, and the resulting solution was stirred at room temperature for 1 h before the sulfur ylide **2b** (310.0 mg, 1.00 mmol, 1.0 equiv.) dissolved in dry DCM (0.5 mL) was added dropwise via a syringe. The reaction was stirred at room temperature for 5 min, and a solution of 2-diazo-*N*,*N*-dimethyl-3-(triisopropylsiloxy)but-3-enamide (**1a**, 373.2 mg, 1.20 mmol, 1.2 equiv.) in dry DCM (1.0 mL) was then added dropwise at rt for 1 h. The reaction mixture was stirred at the same temperature for 16 h, filtered through a short pad of silica gel and washed. Then the reaction solution was filtered through a short pad of Celite. The filtrate was concentrated, and the residue was purified by flash chromatography on silica gel using 10:3 hexanes:ethyl acetate as the eluent to afford (*R*)-4-benzoyl-*N*,*N*-dimethyl-2-(triisopropylsiloxy)cyclobut-1-ene-1-carboxamide (**3a**) as a colorless liquid. 79% yield, 315 mg, 60% ee, HPLC conditions for determination of enantiomeric excess: Chiralpak OD-H column, 254 nm, hexanes/*i*-PrOH = 97:3, 1.1 mL/min, t_r_ (1) = 13.00 min, t_r_ (2) = 14.35 min; ^1^H-NMR (500 MHz, CDCl_3_) δ 8.12–8.09 (m, 2H), 7.55 (tt, *J* = 7.3, 2.2 Hz, 1H), 7.50–7.45 (m, 2H), 4.51 (dd, *J* = 5.0, 1.9 Hz, 1H), 3.23 (s, 3H), 3.94 (s, 3H), 2.88 (dd, *J* = 13.3, 5.0 Hz, 1H), 2.77 (dd, *J* = 13.3, 1.9 Hz, 1H), 1.26–1.18 (m, 3H), 1.15–1.08 (comp, 18H); ^13^C-NMR (125 MHz, CDCl_3_) δ 200.6, 163.5, 147.9, 136.7, 132.9, 128.5, 128.4, 111.7, 38.8, 37.7, 36.1, 34.8, 17.6, 12.5 ppm; HRMS (ESI) *m*/*z* calcd for C_23_H_35_NO_3_Si: [M + H]^+^ 416.2459; found: 416.2448.

### 3.5. Preparation of 4-Benzoyl-N,N-Dimethyl-2-(triisopropylsiloxy)cyclobut-1-ene-1-carboxamide (Racemic ***3a***)

To an oven-dried sealable 2-dram vial equipped with a stir bar were added CuOTf·Tol_1/2_ (4.6 mg, 0.018 mmol, 5 mol%) and sulfur ylide **2b** (91.1 mg, 0.36 mmol, 1.0 equiv.). After the vial was evacuated and backfilled with N_2_ three times, dry DCM (2.0 mL) was added via a syringe; and the solution was stirred at room temperature for 5 min before 2-diazo-*N*,*N*-dimethyl-3-(triisopropylsiloxy)but-3-enamide (**1a**, 127.1 mg, 0.36 mmol, 1.2 equiv.) dissolved in dry DCM (2.0 mL) was added dropwise via a syringe for 1 h. The reaction solution was stirred at room temperature for 16 h then filtered through a short pad of Celite and washed with DCM. The filtrate was concentrated, and the residue was purified by flash chromatography on silica gel using 10:3 hexanes:ethyl acetate as the eluent to afford the corresponding racemic [3 + 1]-cycloaddition product 4-benzoyl-*N*,*N*-dimethyl-2-(triisopropylsiloxy)cyclobut-1-ene-1-carboxamide (Racemic **3a**) as a colorless liquid (115 mg, 87%). Enantiomer composition was determined by HPLC analysis [Daicel chiralpak OD-H, hexanes/*i*-PrOH = 97/3, 1.0 mL/min, λ = 254 nm, t1 = 13.90 min, t2 = 15.69 min.

### 3.6. Preparation of 2-Diazo-N,N-dimethyl-3-(triisopropylsiloxy)but-3-enamide (***1a***)

To a 100 mL oven-dried round bottom flask containing a magnetic stirring bar, commercially available 2-diazo-*N*,*N*-dimethyl-3-oxobutanamide (**6**, 1.0 equiv., 5.18 g, 33.41 mmol) and Et_3_N (1.5 equiv., 6.97 mL, 50.12 mmol) in DCM (100 mL) were added TIPSOTf (1.1 equiv., 9.87 mL, 36.75 mmol) slowly at 0 ℃. After the reaction mixture was stirred for 30 min, hexane (150 mL) was added, followed by saturated aqueous NaHCO_3_ solution (40 mL). The organic phase was separated and washed two more times with saturated aqueous NaHCO_3_ solution (40 mL × 2) then dried with anhydrous Na_2_SO_4_. After evaporating the solvents, the residue was then purified by flash chromatography (SiO_2_ was treated with hexanes with 5% Et_3_N for 10 min before use, hexanes 100% then 10:1 hexanes; ethyl acetate as the eluent) to afford 10.01 g (95%) of 2-diazo-*N*,*N*-dimethyl-3-(triisopropylsiloxy)but-3-enamide (**1a**) as yellow color liquid; ^1^H-NMR (500 MHz, CDCl_3_) δ 4.45 (d, *J* = 2.2 Hz, 1H), 4.27 (d, *J* = 2.2 Hz, 1H,), 3.00 (s, 6H), 1.31–1.21 (m, 3H), 1.14–1.08 (comp, 18H); ^13^C-NMR (125 MHz, CDCl_3_) δ 165.0, 143.8, 89.6, 37.4, 17.7, 17.4, 12.6, 12.3 ppm; HRMS (ESI) *m*/*z* calculated for C_15_H_27_N_3_O_2_Si: [M + H]^+^ 312.2102; found 312.2096.

### 3.7. General Procedure for Catalytic [3 + 1]-Cycloaddition Reaction of Sulfur Ylides (***2b***) with ***1b*** or Cyclopropenecarboxamides (***5***)

#### 3.7.1. General Procedure for Asymmetric Catalytic CuOTf-Catalyzed [3 + 1]-Cycloaddition Reactions to Prepare ***Z*-3b** and ***E*-3b** from **1b** (***Z***:***E*** = 1:3) or **1b** (***Z***:***E*** = 25:1)

To an oven-dried sealable 2-dram vial equipped with a stir bar were added CuOTf·Tol_1/2_ (2.6 mg, 0.010 mmol, 5 mol%) and bisoxazoline ligand **L14** (6.1 mg, 0.012 mmol, 6 mol%). After the vial was evacuated and backfilled with N_2_ three times, dry DCM (1.0 mL) was added via a syringe and the resulting solution was stirred at room temperature for 1 h before sulfur ylide **2b** (60.8 mg, 0.20 mmol, 1.0 equiv.) dissolved in dry DCM (0.5 mL) was added dropwise via a syringe. The reaction was stirred at room temperature for 5 min and then a solution of cyclopropenecarboxamide **1b** (78.0 mg, 0.24 mmol, 1.2 equiv.) in dry DCM (1.0 mL) was then added dropwise for 1 h. The reaction mixture was stirred at the same temperature for 24 h, filtered through a short pad of silica gel and washed with DCM. The filtrate was further concentrated and directly subjected to analysis by ^1^H-NMR. After that, the residue was purified by flash chromatography on silica gel using 10:0 to 10:3 hexanes:ethyl acetate as the eluent to afford the expected [3 + 1]-cycloaddition product ***Z*-3b**, ***E*-3b** and diene (**4**).

For **1b** (***Z***:***E***-**1b** = 1:3) using Ligand **L14**, Ylide **2b** at rt, 24 h: Scale 0.2 mmol, dr 6:1 (***Z***-**3b**:***E***-**3b**), 47 mg of ***Z*****-3b** as colorless liquid, 57% yield, 64% ee; 10% of ***E*-3b**, 7% ee and 15 mg (22%) of **4** as colorless liquid; HPLC conditions for determination of enantiomeric excess of ***Z*-3b**: Chiralpak OD-H column, 254 nm, hexanes/*i*-PrOH = 97:3, 1.0 mL/min, tr (minor) = 13.15 min, tr (major) = 17.19 min; HPLC conditions for determination of enantiomeric excess of ***E*-3b**: Chiralpak OD-H column, 254 nm, hexanes/*i*-PrOH = 97:3, 1.0 mL/min, tr (minor) = 11.31 min, tr (major) = 9.63 min.

For **1b** (***Z***:***E***-**1b** = 25:1) using Ligand **L14**, Ylide **2b** at rt, 24 h: Scale 0.2 mmol, dr 9:1 (***Z*****-3b**:***E*-3b**), 56 mg of ***Z*****-3b** as colorless liquid, 69% yield, 68% ee; 8% of ***E*-3b**, 20% ee and 10 mg (15%) of **4** as colorless liquid; HPLC conditions for determination of enantiomeric excess of ***Z*-3b**: Chiralpak OD-H column, 254 nm, hexanes/*i*-PrOH = 97:3, 1.0 mL/min, tr (minor) = 13.61 min, tr (major) = 18.10 min; HPLC conditions for determination of enantiomeric excess of ***E*-3b**: Chiralpak OD-H column, 254 nm, hexanes/*i*-PrOH = 97:3, 1.0 mL/min, tr (minor) = 11.34 min, tr (major) = 9.61 min.

#### 3.7.2. General Procedure for Asymmetric Catalytic [3 + 1]-Cycloaddition to Prepare ***Z*-3b** and ***E*-3b** from **5**

To an oven-dried sealable 2-dram vial equipped with a stir bar were added CuOTf·Tol_1/2_ (2.6 mg, 0.010 mmol, 5 mol%) and bisoxazoline ligand **L14** (6.1 mg, 0.012 mmol, 6 mol%). After the vial was evacuated and backfilled with N_2_ three times, dry DCM (1.0 mL) was added via a syringe and the resulting solution was stirred at room temperature for 1 h before sulfur ylide **2b** (61 mg, 0.20 mmol, 1.0 equiv.) dissolved in dry DCM (0.5 mL) was added dropwise via a syringe for 1 h. The reaction was stirred at room temperature for 5 min and then a solution of cyclopropenecarboxamide **5** (71 mg, 0.24 mmol, 1.2 equiv.) in dry DCM (0.5 mL) was then added dropwise. The reaction mixture was stirred at the same temperature for 24 h, filtered through a short pad of silica gel and washed with hexanes/EtOAc (1:1, 10 mL). The filtrate was further concentrated and directly subjected to analysis by ^1^H-NMR. After that, the residue was purified by flash chromatography on silica gel using 10:0 to 10:3 hexanes:ethyl acetate as the eluent to afford to afford the expected [3 + 1]-cycloaddition product ***Z*-3b**, ***E*-3b** and diene (**4**).

For **5** using Ligand **L14**, Ylide **2b** at rt, 24 h: Scale 0.2 mmol, dr 12:1 (***Z*****-3b**:***E*-3b**), 56 mg of ***Z*****-3b** as colorless liquid, 67% yield, 71% ee; 5.5% of ***E*-3b**, 16% ee and 5.5 mg (8%) of **4** as colorless liquid; HPLC conditions for determination of enantiomeric excess of ***Z*-3b**: Chiralpak OD-H column, 254 nm, hexanes/*i*-PrOH = 97:3, 1.0 mL/min, tr (minor) = 13.65 min, tr (major) = 17.76 min; HPLC conditions for determination of enantiomeric excess of ***E*-3b**: Chiralpak OD-H column, 254 nm, hexanes/*i*-PrOH = 97:3, 1.1 mL/min, tr (minor) = 11.50 min, tr (major) = 9.74 min.

*(3R,4S)-4-Benzoyl-3-methyl-N,N-dimethyl-2-(triisopropylsiloxy)cyclobut-1-ene-1-carboxamide* (***Z*-3b**): ^1^H-NMR (500 MHz, CDCl_3_) δ 7.94 (d, *J* = 7.5 Hz, 2H), 7.54 (t, *J* = 7.5 Hz, 1H), 7.44 (t, *J* = 7.5 Hz, 2H), 4.64 (d, *J* = 4.9 Hz, 1H), 3.39–3.33 (m, 1H), 3.29 (s, 3H), 2.97 (s, 3H), 1.23–1.05 (comp, 21H), 0.97 (d, *J* = 6.9 Hz, 3H); ^13^C-NMR (125 MHz, CDCl_3_) δ 199.4, 164.3, 150.3, 137.2, 132.9, 128.5, 128.2, 110.6, 44.5, 43.0, 37.9, 34.7, 17.5, 13.0, 12.6; HRMS (ESI) *m*/*z* calcd for C_24_H_37_NO_3_Si: [M + H]^+^ 416.2615; found: 416.2610.

*(3R,4R)-4-Benzoyl-3-methyl-N,N-dimethyl-2-(triisopropylsiloxy)cyclobut-1-ene-1-carboxamide* (***E*-3b**): ^1^H-NMR (500 MHz, CDCl_3_) δ 8.06 (d, *J* = 7.4 Hz, 2H), 7.58–7.54 (m, 1H), 7.48 (t, *J* = 7.4 Hz, 2H), 4.01 (d, *J* = 1.4 Hz, 1H), 3.21 (brs, 3H), 3.01–2.95 (m, 1H), 2.93 (brs, 3H), 1.42 (d, *J* = 7.1 Hz, 3H), 1.23–1.05 (comp, 21H).

*N,N-Dimethyl-3-triisopropylsiloxy-2,4-pentadienecarboxamide* (**4**): ^1^H-NMR (500 MHz, CDCl_3_): δ 6.20 (dd, *J* = 17.1, 10.7 Hz, 1H), 5.64 (dd, *J* = 17.1, 1.2 Hz, 1H), 5.32 (s, 1H), 5.24 (dd, *J* = 10.7, 1.2 Hz, 1H), 3.06 (s, 3H), 2.97 (s, 3H), 1.24–1.18 (comp, 3H), 1.16–1.08 (comp, 18H); ^13^C-NMR (125 MHz, CDCl_3_) δ 167.3, 153.3, 135.3, 117.0, 37.9, 34.6, 17.9, 13.5; HRMS (ESI) *m*/*z* calcd for C_16_H_31_NO_2_Si: [M + H]^+^ 298.2197; found: 298.2193.

### 3.8. Preparation of 2-Diazo-N,N-dimethyl-3-oxopentanamide (***8***)

To a stirred solution of *N*,*N*-dimethyl-3-oxopentanamide [35] (**7**, 3.00 g, 21.0 mmol) and *p*-acetamidobenzenesulfonyl azide (*p*-ABSA, 5.00 g, 21.0 mmol) in acetonitrile (80 mL), Et_3_N (8.75 mL, 63.0 mmol) was added dropwise at 0 °C over 3 min. The reaction mixture was warmed to room temperature and stirred for 16 h. Acetonitrile was then removed under reduced pressure, and the residue was redissolved in dichloromethane. The sulfonamide precipitate was filtered, and the filtrate was concentrated under reduced pressure. The residue was then purified by silica gel column chromatography using a 10:4 to 10:6 gradient of hexane/ethyl acetate (*v*/*v*) as the eluent to afford 2.95 g (83%) of 2-diazo-*N*,*N*-dimethyl-3-oxopentanamide (**8**) as yellow color liquid. ^1^H-NMR (500 MHz, CDCl_3_) δ 3.00 (s, 6H), 2.65 (q, *J* = 7.3 Hz, 2H), 1.14 (t, *J* = 7.3 Hz, 3H); ^13^C-NMR (125 MHz, CDCl_3_) δ 192.7, 161.3, 37.4, 32.4, 8.3 ppm; HRMS (ESI) *m*/*z* calculated for C_7_H_11_N_3_O_2_: [M + H]^+^ 170.0924; found 170.0923.

### 3.9. Preparation of 2-Diazo-N,N-dimethyl-3-((triisopropylsiloxy)pent-3-enamide, ***1b*** (**Z**:**E-1b** = 1:3)

**1b** (***Z***:***E*** = 1:3) was prepared from 2-diazo-*N*,*N*-dimethyl-3-oxopentanamide (**8**) according to reported [18,32] procedure using TIPSOTf (1.1 equiv.), Et_3_N (1.5 equiv.) and LiCl (1.0 equiv.). Its ***Z***/***E*** ratio was determined by ^1^H-NMR spectroscopy and 10% formation of cyclopropene **5** was observed compared to total olefin (***E*** + ***Z***). For simplicity only ^1^H-NMR of ***E*** isomer was given here; yield 95%; ^1^H-NMR of ***E*** (500 MHz, CDCl_3_) δ 4.93 (q, *J* = 7.1 Hz, 1H), 2.99 (s, 6H), 1.57 (d, *J* = 7.1 Hz, 3H), 1.32–1.02 (m, 21H); HRMS (ESI) *m*/*z* calculated for C_16_H_31_N_3_O_2_Si: [M + H]^+^ 326.2258; found 326.2253.

### 3.10. Preparation of 2-Diazo-N,N-dimethyl-3-(triisopropylsiloxy)pent-3-enamide, ***1b*** (**Z**:**E-1b** = 25:1)

The reported [30] procedure was modified to make 2-diazo-*N*,*N*-dimethyl-3-((triisopropylsiloxy)pent-3-enamide (***Z***:***E*-1b** = 25:1). The title compound includes a small percent (5–10%) of cyclopropenecarboxamide **5** because enoldiazoacetamide undergoes dinitrogen extrusion to form cyclopropenecarboxamide, even when stored at 0 °C. Compound **1b** (***Z***:***E*** = 25:1) in common NMR solvents (CDCl_3_, CD_2_Cl_2_, DMSO_d6_, CD_3_NO_2_ and hexanes) is converted (at 0 °C) to the corresponding cyclopropene (**5**), but was found to be stable at −20 °C and should be stored at −20 °C. At 0 °C and rt, 20% and 70%, formation of **5** was observed after 6 h respectively.

To a 100-mL oven-dried vial equipped with a magnetic stirring bar, 2-diazo-*N*,*N*-dimethyl-3-oxopentanamide (**8**, 200 mg, 1.18 mmol) was added, and the system was filled with nitrogen. THF (30 mL) was then added, and the reaction solution was cooled to −78 °C (dry ice/acetone bath), LiHMDS (1.30 mL, 1.0 M in the hexanes) was introduced dropwise over 2 min, followed by the addition of TIPSOTf (0.315 mL, 1.18 mmol) at −78 °C. The resulting solution was stirred at −78 °C until the reaction was complete (monitored by TLC, about 5−15 min). The reaction was quenched at −78 °C with ice-cold brine, and solvent was removed at 15 °C under reduced pressure, and the residue was directly purified by column chromatography on silica gel which was pre-treated with 5 vol.% triethylamine/hexanes (eluent: pure hexanes then 10% EtOAc in hexanes) to give the desired product 2-diazo-*N*,*N*-dimethyl-3-(triisopropylsiloxy)pent-3-enamide, **1b** (***Z***:***E*-1b** = 25:1) as yellow liquid which was immediately stored −20 °C. ^1^H-NMR of **1b-*Z*** (500 MHz, CDCl_3_) δ 5.00 (q, *J* = 6.9 Hz, 1H, Z-H), 3.01 (s, 6H, NCH_3_), 1.70 (d, *J* = 6.9 Hz, 3H, CH_3_), 1.32–1.02 (m, 21H, TIPS); ^13^C-NMR of **1b-*Z*** including **5** (125 MHz, CDCl_3_) δ 160.7, 142.5, 79.6, 37.1, 34.6, 34.5, 31.5, 25.8, 25.2, 22.6, 20.3, 17.7, 17.4, 17.4, 14.1, 12.3, 12.0, 11.8 ppm; HRMS (ESI) *m*/*z* calculated for C_16_H_31_N_3_O_2_Si: [M + H]^+^ 326.2258; found 326.2254.

### 3.11. Preparation of N,N,3-Trimethyl-2-(triisopropylsiloxy)cycloprop-1-ene-1-carboxamide (***5***)

2-Diazo-*N*,*N*-dimethyl-3-(triisopropylsiloxy)pent-3-enamide, **1b** (***Z***:***E*-1b** = 1:3) (1.00 g, 3.17 mmol) at room temperature was added to a 20 mL oven dried vial containing a magnetic stirring bar and 10 mL CHCl_3_, and then the vial was screwed close. The sealed reaction mixture was heated in oil bath at 50 °C for 2 h, during this time the diazo compound was converted to the corresponding cyclopropene. The color of the solution changed from yellow to colorless. Solvent was removed under reduced pressure to afford 980 mg (>99%) of *N*,*N*,3-trimethyl-2-(triisopropylsiloxy)cycloprop-1-ene-1-carboxamide (**5**) was obtained as colorless liquid and was characterized directly without further purification. Compound **5** was also obtained heating **1b** (***Z***:***E*-1b** = 25:1) in CHCl_3_ at 50 °C for 1 h. ^1^H-NMR (500 MHz, CDCl_3_) δ 3.16 (s, 3H), 2.99 (s, 3H), 2.38 (q, *J* = 4.9 Hz, 1H), 1.30 (d, *J* = 4.9 Hz, 3H), 1.36–1.24 (comp, 3H), 115–1.10 (comp, 18H); ^13^C-NMR (125 MHz, CDCl_3_) δ 160.7, 142.5, 79.6, 37.1, 34.5, 25.8, 20.3, 17.9, 17.8, 17.7, 17.7, 17.5, 17.4, 17.4, 12.3, 12.0 ppm; HRMS (ESI) *m*/*z* calculated for C_16_H_31_NO_2_Si: [M + H]^+^ 298.2197; found 298.2193.

## 4. Conclusions

The influence of ligands on stereocontrol with γ-methyl-substituted **1b** is surprisingly diverse. Diastereoselectivity favoring the *cis* isomer ranges from 2:1 with SaBox ligand **L1** (Table 2, entry 1) to >20:1 with the tetramethylene SaBox ligand **L14** (Table 2, entry 7) (the *cis*-4,5-diphenyl-Box template) providing the highest diastereocontrol. The highest enantioselectivity is also found with **L14** (Table 2, entries 6, 7) which appears to be the optimum between smaller (**L12**, Table 2, entry 5) and larger (**L16** and **L17**, Table 2, entries 8, 9) attachments. Similarly, the *cis*-4,5-diphenyl-Box template of **L14** (Table 2, entry 6) provides higher stereocontrol than the analogous indanyl-Box template of **L11** (Table 2, entry 4). For [3 + 1]-cycloaddition with *N,N*-dimethyl-enoldiazoacetamide **1a** ligand **L14** (Table 1, entry 7) exhibited much lower enantioselectivity, 34% ee compared to 60% ee with **L17** (Table 1, entry 10), demonstrating a significant dependence of enantioselectivity on the chiral ligand. However, enantioselctivity for formation of the *trans* donor-acceptor cyclobutene isomer ***E*-3b** was very low in comparison and exhibited no obvious means for improvement with the ligands that were employed. Attempted extension of this methodology to the γ-ethyl-substituted enoldiazoacetamide using the same copper catalyst with ligand **L14** at room temperature gave low product yield, low diastereoselectivity (2:1 *cis*:*trans*) and low enantioselectivity, which further signaled the inherent complexity of stereocontrol in these reactions.

The competitive formation of diene **4** was another surprise in this transformation. Not previously observed from reactions with the corresponding enoldiazoacetates [18], this product was observed in variable amounts from all reactions catalyzed by copper(I) with Box or Sabox ligands. However, diene **4** (Appendix A) was not formed when the copper catalyzed reaction with ylide **2b** was performed without ligand with either enoldiazoacetamide ***Z*-1b** or with donor-acceptor cyclopropenecarboxamide **5**, and neither the sulfur ylide nor the ligand alone caused diene formation in reactions performed over 24 h. Since a 20% excess of ligand was used to ensure that unligated copper(I) was minimized, we thought that the diene might arise from the ligand, acting as a base, to effect elimination from the intermediate metallo-enol carbene. Indeed, using 2.4 mol% triethylamine in place of the ligand resulted in diene formation (29% yield, Appendix A). However, limiting the amount of excess ligand in the [3 + 1]-cycloaddition reaction to exactly 1:1 correspondence with the copper catalyst did not reduce the amount of diene **4** formation; but lowering the temperature to −20 °C brought diene formation down to 2%.

In summary, Cu(I) catalyzed asymmetric [3 + 1] cycloaddition of α-benzoyl diphenylsulfur ylide **2b** with 3-methyl- or un-substituted cyclopropenecarboxamides gives access to the synthesis of donor-acceptor cyclobutenecarboxamides in good yield and moderate enantioselectivity. Unlike their corresponding enoldiazoacetate, the γ-methyl substituted amide gives a high preference for the *cis* diastereoisomer. Reactivity and stereoselectivity of the amide and ester are significantly different, and formation of diene **4** from amide **1b** suggests an elimination pathway for the intermediate metallo-enolcarbene.

## Data Availability

Not applicable.

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
