# Peer review of "Challenges in the Highly Selective [3 + 1]-Cycloaddition of an Enoldiazoacetamide to Form a Donor–Acceptor Cis-Cyclobutenecarboxamideâ€"

_molecules, 2021, doi:10.3390/molecules26123520_

Round 1
Reviewer 1 Report
The manuscript submited by Doyle and co-workers describes the efforts to achieve diastereo- and enantio-selective [3+1]-cycloaddition of an 2-enoldiazoacetamide to a donor–acceptor cyclopropenecarbox-3 amide in the synthesis of chiral cyclobutenes. The paper is well-written and full of experiments, even though stereoselectivity is only moderate. Therefore, publication is highly recommended. I would improve the manuscript citing more precisely in the text the entry of the respective tables. The supplementary material shows some repetitive spectra as for 3a, which should be removed.
Author Response
We thank this reviewer for these comments. We have cited more precisely the entry of the respective tables throughout the manuscript and removed the repetitive spectral descriptions from the Supplementary Information.
Reviewer 2 Report
According to my opinion, the manuscript entitled "Challenges in the Highly Selective [3+1]-Cycloaddition of an Enoldiazoacetamide to a Donor-Acceptor Cyclopropenecarboxamide" given by Sipak Joyasawa, Donghui Ma and Michael P. Doyle, is appropriate for Molecules. I agree with the Authors that the ee is moderate, it is an essential drawback of this work. However, the results are promising and maybe in the future examinations, higher ee can be achieved. I expect that NMR spectra of some products (sup info, page 8, 9, 10, 12, 13, 14, 15, 17, 18, 20) will be corrected, namely the intensity of some signals will be higher and selected spectrum fragments will be widened. I am afraid that the purity of some compounds has to be corrected. Additionally, structures on the spectra should be correlated with the spectra: appropriate protons should be indicated in the figures and in the spectra. In the current form, the spectra are really substandard and cannot be accepted by me.
Author Response
We thank this reviewer for these comments. As suggested, the intensity of signals in the denoted NMR spectra has been increased and spectral fragments have been added to better recognize coupling. We have also correlated absorptions with several of the spectra. There is one compound (1b) whose purity is in question, and the reasons for this are explained in the manuscript. This compound is prone to form the donor-acceptor cyclopropene 5 at room temperature, and this transformation is documented in the SI.
Reviewer 3 Report
This manuscript is devoted to the study of the asymmetric cycloaddition of dimethyl or diphenylsulphonium α-benzoylmethylides to 2-diazo-N,N-dimethyl-3-(triisopropylsiloxy)but-3-enamide and the corresponding pent-3-enamide. This manuscript can be published in Molecules after revision.
Firstly, title of manuscript does not correspond its content. Authors did not studied (3+1)-Cycloaddition of enoldiazoacetamide to a donor-acceptor cyclopropenecarboxamide. Therefore, title should be changed. In general, it seems that Prof. Doyle did not read this manuscript before sending to the journal due to multiple misprint and errors in the text.
In particular, in Scheme 1 (b and c sections) text “Catalyst control of both diastereo- enantio-selectivity” should be changed by “Catalyst control of both diastereo- and enantio-selectivity”.
The terms a-acyldimethylsulfur ylide and α-acyldiphenylsulfur ylide are ambiguous; it would be better to use dimethyl or diphenylsulphonium a-benzoylmethylides.
Descriptors Z- and E- are recommended by IUPAC to use for the presentation of the relative stereochemistry at double bonds only; for ring compounds cis- and trans- are preferable.
Yields and enantioselectivitis in Scheme 3 differ from the data given in the text and in Experimental Section.
In table 2 “Yield ee (Z-3b)” and “Yield ee (E-3b)” should be changed by “Yield (ee) of Z-3b” and “Yield (ee) of E-3b”, respectively (if we will ignore the IUPAC recommendation on the use of Z- and E-desciptors).
Line 237: 0.06 mmol, 6 mmol %
Lines 262,263 and lines 291,292: 0.010 mmol, 5 mol% but 0.012 mmol, 1.0 equiv.
Lines 270,271 and lines 299,300: “filtrate was further concentrated and directly subject”
Line 272. “to afford to afford”
Lines 310 and 316: Titles for compounds of Z-3b and E-3b are different despite they are diastereomers only.
Line 318. Brss. At whole, the description of E-3b is not good. Firstly, 13C spectral data are desirable. Secondly, H(3) atom of cyclobutene should appear as multiplet due to the presence of coupling constants with both the vicinal endocyclic proton and exocyclic methyl group. But this signal is described as broad singlet. This is either error or the result of bad recording the PMR spectrum.
Line 323. Two signals in 13C NMR spectrum are given as 17.9. By the way, coupling constants for cis-protons at C4 and C5 atoms in 4 are given as 10.7 Hz and 9.8 Hz. These values should be the same for two protons.
Moreover, the explanation of the different behaviour of the related esters and amides is highly desirable.
Some recommendations of the journal for manuscript presentation are ignored but I believe that editor will point out these problems.
At last, some parts of text can be given in more clear und understandable manner.
To conclude, I recommend for accepting this manuscript only after careful revision of the text by authors.
Author Response
We are grateful to the reviewer for the detailed comments. The title of the manuscript was missing a key word and this word has been added.
In particular, in Scheme 1 (b and c sections) text “Catalyst control of both diastereo- enantio-selectivity” should be changed by “Catalyst control of both diastereo- and enantio-selectivity”.
These changes have been made.
The terms a-acyldimethylsulfur ylide and α-acyldiphenylsulfur ylide are ambiguous; it would be better to use dimethyl or diphenylsulphonium a-benzoylmethylides.
We appreciate the reviewer’s attention to this terminology, but a review of current use of the ylide designation is pronounced in the literature and is the same as our prior descriptions (references 18 and 31). Additionally, there should be no ambiguity about their identity since their chemical structures are provided in multiple places in the manuscript.
Descriptors Z- and E- are recommended by IUPAC to use for the presentation of the relative stereochemistry at double bonds only; for ring compounds cis- and trans- are preferable.
The use of cis and trans for ring compounds has been made.
Yields and enantioselectivitis in Scheme 3 differ from the data given in the text and in Experimental Section.
The data has been corrected.
In table 2 “Yield ee (Z-3b)” and “Yield ee (E-3b)” should be changed by “Yield (ee) of Z-3b” and “Yield (ee) of E-3b”, respectively (if we will ignore the IUPAC recommendation on the use of Z- and E-desciptors).
This change has been made.
Line 237: 0.06 mmol, 6 mmol %
Lines 262,263 and lines 291,292: 0.010 mmol, 5 mol% but 0.012 mmol, 1.0 equiv.
Lines 270,271 and lines 299,300: “filtrate was further concentrated and directly subject”
Line 272. “to afford to afford”
Lines 310 and 316: Titles for compounds of Z-3b and E-3b are different despite they are diastereomers only.
These changes have been made.
Line 318. Brss. At whole, the description of E-3b is not good. Firstly, 13C spectral data are desirable. Secondly, H(3) atom of cyclobutene should appear as multiplet due to the presence of coupling constants with both the vicinal endocyclic proton and exocyclic methyl group. But this signal is described as broad singlet. This is either error or the result of bad recording the PMR spectrum.
The H(3) atom of cyclobutene has been designated as a multiplet. We put enormous efforts into the isolation of the pure minor trans diastereomer. However, the trans isomer could not be fully separated from the cis-isomer. The proton spectrum of the mixture was used to identify the trans isomer whose absorptions correspond to the previously well characterized trans isomer of the corresponding ester (reference 18).
Line 323. Two signals in 13C NMR spectrum are given as 17.9. By the way, coupling constants for cis-protons at C4 and C5 atoms in 4 are given as 10.7 Hz and 9.8 Hz. These values should be the same for two protons.
These corrections have been made.
Moreover, the explanation of the different behaviour of the related esters and amides is highly desirable.
With changes made in this manuscript, we believe that the different behavior of corresponding esters and amides are explained in sufficient detail.
Some recommendations of the journal for manuscript presentation are ignored but I believe that editor will point out these problems.
Additions have been made.
At last, some parts of text can be given in more clear und understandable manner.
We have added several phrases that we hope will increase understanding.